# Impact of Salinity on the Energy Transfer between Pigment–Protein Complexes in Photosynthetic Apparatus, Functions of the Oxygen-Evolving Complex and Photochemical Activities of Photosystem II and Photosystem I in Two *Paulownia* Lines

**DOI:** 10.3390/ijms24043108

**Published:** 2023-02-04

**Authors:** Martin A. Stefanov, Georgi D. Rashkov, Ekaterina K. Yotsova, Anelia G. Dobrikova, Emilia L. Apostolova

**Affiliations:** Institute of Biophysics and Biomedical Engineering, Bulgarian Academy of Sciences, Acad. G. Bonchev Str., Bl. 21, 1113 Sofia, Bulgaria

**Keywords:** electron transport, low-temperature chlorophyll fluorescence, thylakoid membranes, NaCl treatment

## Abstract

The present study shows the effect of salinity on the functions of thylakoid membranes from two hybrid lines of *Paulownia*: *Paulownia tomentosa* x *fortunei* and *Paulownia elongate* x *elongata*, grown in a Hoagland solution with two NaCl concentrations (100 and 150 mM) and different exposure times (10 and 25 days). We observed inhibition of the photochemical activities of photosystem I (DCPIH_2_ → MV) and photosystem II (H_2_O → BQ) only after the short treatment (10 days) with the higher NaCl concentration. Data also revealed alterations in the energy transfer between pigment–protein complexes (fluorescence emission ratios F_735_/F_685_ and F_695/_F_685_), the kinetic parameters of the oxygen-evolving reactions (initial S_0_-S_1_ state distribution, misses (α), double hits (β) and blocked centers (S_B_)). Moreover, the experimental results showed that after prolonged treatment with NaCl *Paulownia tomentosa* x *fortunei* adapted to the higher concentration of NaCl (150 mM), while this concentration is lethal for *Paulownia elongata* x *elongata*. This study demonstrated the relationship between the salt-induced inhibition of the photochemistry of both photosystems and the salt-induced changes in the energy transfer between the pigment–protein complexes and the alterations in the Mn cluster of the oxygen-evolving complex under salt stress.

## 1. Introduction

In recent years, an increase in the area of saline soil has become an environmental problem worldwide [1,2]. Salinity causes a physiological and metabolic imbalance in plants as a result of the disruption of enzyme structures, membrane integrity, and cellular metabolism [3], which in turn affects plant reproduction, germination, development, photosynthesis, respiration, and yield [4,5,6,7,8]. During the adaptation of plants in conditions of increased salinity, various mechanisms and processes associated with changes in their morphology, anatomy, and physiology are activated [9,10,11,12]. The mechanisms of salt tolerance include both changes related to the course of individual biochemical pathways, and more complex mechanisms leading to the modification of the activities of key processes such as photosynthesis and respiration, as well as to changes in the processes related to water uptake, transpiration, and stomatal activity [4,13,14]. Salt stress also enhances the generation of reactive oxygen species (ROS) and leads to cellular damage [4,14,15,16,17]. The salt-induced effects in plants can vary depending on climatic conditions, stress duration, plant species, and/or soil conditions [18,19].

Among the various metabolic processes that are affected by salt stress, photosynthesis is one of the most sensitive [8,20,21,22]. When applied over several days, salinity induces ionic stress, affecting photosynthetic efficiency [14,23,24], while osmotic stress occurs in the initial stages of high salt exposure, reversibly slowing down the photosynthetic rate [25,26]. Additionally, osmotic stress affects photosystem II (PS II) antenna heterogeneity, while ionic stress-induced changes occur in both antenna size heterogeneity and reducing side heterogeneity [20,25]. These changes in PSII heterogeneity may be part of adaptation mechanisms activated under high salinity conditions [25].

Salt-induced structural changes also affect the interactions of chlorophyll and protein molecules, leading to instability in pigment–protein complexes [27,28]. Changes have been observed in the proteins of the light-harvesting complex of PSII (LHCII) [29,30,31], oxygen-evolving complex (OEC) [32,33], D1 core protein of PSII [9,34,35] and/or PsaB core subunit of PSI [36]. Modifications of the complexes of the photosynthetic apparatus under salt stress in turn affect functions of the photosynthetic apparatus [37,38]. The opinions of different authors about the influence of salt stress on PSII are contradictory depending on the plant species [7]. Some studies have shown a significant inhibitory effect on PSII activity under salinity [20,39], while others have found no impairment of its structure and function [40,41]. In many experiments conducted with barley, spinach, and sorghum, it has been shown that salinity (100–200 mM NaCl) alone does not affect (or even stimulate) the net photosynthetic rate, preserving PSII functional integrity in treated plants [13,42,43,44]. In experiments with salt-sensitive plants such as rice, peas and tomatoes, a damaging effect of high salt content on the structure and function of PSII has been found, as the inhibition occurs in the electron transport pathway at the PSII donor and acceptor sides [13,45,46,47], as well as on composition and functions of the OEC [38,48,49,50]. In addition, salt stress was found to have a greater impact on the donor side of PSII, while the acceptor side was relatively less sensitive, and some of the salt-induced damage in PSII was reversible after removing the stress. The study of Mehta et al. [46] on wheat plants revealed that the acceptor side is restored 100%, while the donor side is restored only 85% after the termination of the salt treatment.

Chlorophylls (Chl) and carotenoids (Car) are the main photosynthetic pigments that play an important role in photosynthesis [51]. Salt stress can affect the Chl and Car contents, as the changes depend on the amount of salt and the plant species [34,52]. Some authors use the changes in the pigment amount as a sensitive indicator for the salt-sensitivity of the plants [23,53]. In salt-tolerant species, the Chl content was found to increase, while in salt-sensitive species it decreases [54,55].

*Paulownia* is a genus that belongs to the class *Lamiales* (family *Paulowniaceae*). It originates from China and has subsequently cultivated and used widely in Europe [56,57]. These fast-growing tree species with large heart-shaped leaves, a well-developed root system, and the ability to adapt easily and quickly to environmental conditions have been the subject of increased interest in the last decade for conducting research on the application possibilities of different species of the genus *Paulownia* for phytoremediation of contaminated soils [53,56,57,58]. The sensitivity to salinization varies in different *Paulownia* species [59,60,61]. Our recent investigations on the influence of different soil salinities on photosynthetic performance in two hybrid lines of *Paulownia* (*Paulownia tomentosa* x *fortunei* (TF) and *Paulownia elongata* x *elongata* (EE)) using chlorophyll fluorescence measurements have found that the TF line is more tolerant to high soil salinity than the EE line [53,62]. The aim of the present study was to investigate how salt-induced changes in the energy transfer between the pigment–protein complexes and the kinetic parameters of oxygen evolution influence the degree of photochemical inhibition of the two photosystems. This will provide more detailed information on the molecular mechanisms involved in the adaptive responses of the photosynthetic apparatus to salt stress. Thus, we studied isolated thylakoid membranes from seedlings of both hybrid lines (TF and EE) grown hydroponically after short (10 days) and prolonged (25 days) treatment with NaCl (100 mM and 150 mM). The changes in pigment composition, the energy transfer between pigment–protein complexes, the photochemical activities of PSII and PSI, and the kinetic parameters of the oxygen-evolving reactions (i.e., alterations in the Mn cluster of OEC) were investigated. The data obtained from this study will contribute to elucidating some of the reasons for the different tolerance of the two *Paulownia* lines and will provide new information about the mechanisms of plant adaptation and tolerance to salinity, as well as their potential to be used for phytoremediation of saline soils.

## 2. Results

### 2.1. Pigment Composition

The analysis of pigment composition revealed that leaf Chl content in the leaves of untreated plants was greater in the TF than in the EE line. The short treatment (10 days) with 100 and 150 mM NaCl led to a decrease in the Chl *a* and Chl *b* in the EE line from 9% to 20% for Chl *a* and from 10% to 22% for Chl *b*), whereas a decrease in the Chl content of the leaves of the TF line was registered only after treatment with 150 mM NaCl (by 11% for Chl *a* and by 13% for Chl *b*) (Figure 1c,d).

Salt treatment for 10 days led also to a decrease in leaf Car content in both lines, depending on the applied NaCl concentration (Figure 2b). The Car reduction was registered in the EE line after treatment with 100 mM (by 11%) and 150 mM NaCl (by 23%), while in the TF line only after treatment with 150 mM NaCl (by 18%) (Figure 2b). After the prolonged salt treatment (25 days), the Car content in both lines was similar to that of untreated plants (Figure 2a).

Experimental results showed that after the prolonged treatment with NaCl (25 days), the chlorophylls in the TF line were less influenced at 100 and 150 mM NaCl, while the EE line treated with 150 mM NaCl did not survive (Figure 1a,b). Moreover, the Car content in the leaves of TF line was greater than that in the EE line for all the variants studied. This indicates that the TF line has adapted better to high salt stress than the EE line.

### 2.2. Absorption Spectra

The absorption spectra of isolated thylakoid membranes from two *Paulownia* lines (TF and EE) are characterized by typical bands in the blue wavelengths (400–500 nm), due to Chl (*a* and *b*) and carotenoid absorption, as well as bands in the red region (600–700 nm), due to the Chl absorption only [63]. The absorption maxima for *Paulownia* thylakoid membranes were observed at 442 nm and 680 nm for Chl *a*, as well as at 472 and 652 nm for Chl *b* (Figure 3). Compared to untreated plants, the TF thylakoid spectrum after 10 days of treatment with 150 mM NaCl showed smaller amplitudes at 442 nm and 472 nm by about 15% and 10%, respectively (Figure 3a). At the same time, this decrease in the EE thylakoids was by about 25% at 442 nm and 472 nm (Figure 3b). In addition, the spectra of the thylakoid membranes from both *Paulownia* lines treated with 150 mM NaCl revealed also a decrease in the absorption maxima in red wavelengths (652 nm and 680 nm), as this decrease was smaller in TF (by 11%) than in EE thylakoids (by 24%) (Figure 3a,b). The absorption spectra of thylakoid membranes from both lines after prolonged (25 days) salt treatment were similar to the respective controls (Appendix A).

### 2.3. Low-Temperature Chlorophyll Fluorescence

Low-temperature (77 K) chlorophyll fluorescence spectra were used to estimate the energy transfer between pigment–protein complexes of the photosynthetic apparatus. The fluorescence emission spectra of the studied thylakoid membranes were characterized by three maxima: at 685 nm and at 695 nm, associated with PSII complex, and one at 735 nm, associated with PSI complex/unit [64]. The fluorescence emission ratio F_735_/F_685_ characterizes the energy redistribution between the PSI and PSII, while the ratio F_695_/F_685_ characterizes the energy transfer between the pigment–protein complexes in PSII. The ratio F_735_/F_685_ (at excitation with 436 nm and 472 nm) increased for both studied lines after short treatment with 150 mM NaCl, while this ratio was similar to the respective controls after treatment with 100 mM NaCl (Table 1). Moreover, the values of this ratio (F_735_/F_685_) were greater in the EE line than the TF line in all the variants studied (Table 1). In addition, data showed that the excitation at 472 nm also led to an increase in the ratio fluorescence emission ratio F_735_/F_685_ after treatment with 150 mM NaCl, but this increase was smaller than after excitation with 436 nm (Table 1). In addition, the ratio F_695_/F_685_ after excitation with 436 nm and 472 nm also increased after treatment with 150 mM NaCl for 10 days in both lines. Furthermore, the values of both fluorescence emission ratios (F_735_/F_685_ and F_695_/F_685_) were similar to those of their respective controls after prolonged treatment with NaCl (25 days) (Table 1).

For more detailed information on the individual contribution of the pigment–protein complexes of thylakoid membranes to the total 77 K chlorophyll emission spectrum, we used the method of decomposition of the spectrum into six main bands as shown in Andreeva et al. [65]. The main bands in *Paulownia* thylakoid membranes had maxima at 680, 685, 695, 700, 720 and 735 nm, respectively attributed to LHCII (trimers and monomers, LHCII^T+M^), the reaction center of the PSII complex (PSII RC), core antenna complex PSII (PSII antenna), LHCII (aggregated trimers, LHCII^A^), core complex of PSI (PSI core) and antenna complex of PSI (PSI antenna). Short treatment (10 days) with 150 mM NaCl led to a decrease in the fluorescence emitted from LHCII^T+M^, PSII core antenna and PSII RC. It should be noted that under these conditions, the ratio LHCII^A^ to LHCII^T+M^ increased more strongly in the EE line (from 1.53 for the control to 2.39) than in TF line (from 0.98 for the control to 1.37) after 150 mM NaCl exposure (Table 2). In addition, data revealed that the fluorescence attributed to the PSI core increased by 43% for the TF line and by 10% for the EE line, while the PSI antenna increased by 8 and 19% for TF and EE line, respectively. On the other hand, the fluorescence attributed to monomers and trimers of LHCII, and the RC of the PSII complex decreased in both studied lines. The short-duration salt stress (10 days) led also to an increase in the fluorescence emission of LHCII^A^ (Table 2).

### 2.4. Photochemical Activity

The photochemical activities of both photosystems were evaluated by the PSI- and PSII- dependent electron transport in isolated thylakoid membranes. Data revealed that these activities were changed to different extents depending on the NaCl concentration and time of treatment (Figure 4). The PSII-dependent electron transport was determined in the presence of an exogenous acceptor benzoquinone (H_2_O → BQ). The results indicated an inhibition of the PSII-dependent electron transport in both studied lines after 10 days of treatment with 150 mM NaCl (Figure 4c), as the inhibition was greater in the EE (by 59%) than in the TF line (by 37%). The rate of PSI-dependent electron transport was also inhibited after short treatment with 150 mM NaCl in both studied lines (Figure 4d). The inhibition was by 21% in the TF line and by 48% in the EE line. After prolonged (25 days) salt treatment, the photochemical activities of PSII and PSI were similar to the respective controls in both investigated lines (Figure 4a,b).

### 2.5. Oxygen Evolution under Flash and Continuous Illumination 

The flash-induced yields and the initial oxygen burst under continuous illumination of the thylakoid membranes isolated from control and NaCl-treated leaves of both studied *Paulownia* lines were determined by using a polarographic oxygen rate electrode (Joliot-type) without an artificial acceptor. Amplitudes of the maximum flash-induced oxygen yield observed after the third flash (*Y_3_*) and of the oxygen burst under continuous illumination (*A*) were assessed. The oxygen burst decay showed biphasic kinetics with fast (A_F_) and slow (A_S_) components (see Appendix A). The ratio A_F_/A_S_ was proposed to correspond to the ratio of the functionally active PSIIα to PSIIβ centers located in grana and stroma lamellae, respectively [66,67]. From the analysis of the flash-induced oxygen yields, the following parameters were also determined: the populations of PSII centers in the most reduced state in darkness (S_0_), misses (α), the double hits (β) and the blocked PSII reaction centers (S_B_). Data showed that after 10 days of treatment the amplitudes of oxygen bursts under continuous illumination (*A*) decreased in the TF line by 20% only after 150 mM NaCl treatment and in the EE line by 15 and 25% after 100 and 150 mM NaCl, respectively (Figure 5d). At the same time, a decrease in the A_F_/A_S_ ratio was also found after short treatment with 150 mM NaCl (Table 3). The values of flash-induced oxygen yields (*Y_3_*) decreased gradually with increasing NaCl concentration, which was more pronounced for the EE than the TF line (Figure 5c). 

The analysis of the flash oxygen yields showed that the amount of PSII centers in the initial S_0_ state, misses (α), and double hits (β) rose for both the TF and EE lines only after 10 days of treatment with 150 mM NaCl (Table 3). The amount of blocked oxygen-evolving PSII centers (S_B_) after the short-duration salt treatment with 100 mM NaCl was found to be changed only in the EE line compared to the control (Table 3). Additionally, an increase in S_B_ was registered in both lines after treatment with the higher concentration of NaCl (150 mM). This increase was of 56% in EE, and 42% in TF compared to the respective controls (Table 3).

Experimental results revealed that after prolonged NaCl exposure (25 days) the values for oxygen burst (*A*) and ratio A_F_/A_S_ were similar to the respective controls except for in the EE line, which did not survive after prolonged treatment with 150 mM NaCl (Figure 5b, Table 3). In addition, a slight inhibition of flash oxygen yields was found in the TF line only after prolonged exposure to 150 mM NaCl, and in the EE line after treatment with 100 mM NaCl (Figure 5a). During 25 days of NaCl treatment, changes in the S_0_–S_1_ distribution in the darkness, misses (α), and double hits (β) were not detected in either of the studied *Paulownia* lines (Table 3). Furthermore, after 25 days of NaCl treatment, no statistically significant changes were registered for the S_B_ parameter in either of the studied lines (Table 3). 

## 3. Discussion

Abiotic environmental stressors, including salinity, strongly affect photosynthesis and inhibit the growth and development of plants [68]. Photosynthetic membranes are especially sensitive to salinity [69,70,71]. The inhibition of the function of the photosynthetic apparatus is connected with structural changes in the thylakoid membranes and modification of their pigment–protein complexes [25,69]. It has been shown that salt concentrations which are harmful to one plant species may not be stressful for another. Previous investigations showed that changes in the proteins of PSII (LHCII, inner antenna of PSII, D1 and OEC) and PSI as well as the degree of the inhibition of photosynthetic performance depend on the salt sensitivity of the plant species [18,19,28].

The results of this study revealed that the NaCl concentrations (100 mM and 150 mM) and the time of treatment have different effects on both *Paulownia* lines (TF and EE). After prolonged treatment (25 days), the concentration of 150 mM NaCl was lethal to the EE plants, while the TF plants adapted better to both tested concentrations (100 mM and 150 mM).

Experimental results showed a reduction in leaf pigments after the short treatment (10 days) in both studied lines (Figure 1 and Figure 2). Analysis of the absorption spectra also revealed smaller salt-induced changes in the absorption bands in the blue and red regions after the short treatment for the TF line than the EE line (Figure 3). Previously, it has been proposed that the decrease in Chl *a* absorption bands is the result of the higher extent of light-scattering by the thylakoid membranes [63]. Therefore, it could be suggested that the higher NaCl concentration during 10 days of exposure causes structural changes in the thylakoid membranes, which were more pronounced in the EE line than in the TF line (Figure 3). Moreover, the absorption spectra of thylakoid membranes from both lines after 25 days of salt treatment were similar to the respective controls, indicating a structural similarity to untreated plants (Appendix A).

In addition, data revealed a stronger decrease in the amount of Chl *a* than Chl *b* in leaves (Figure 1), which could be a result of the influence on pigment biosynthesis and degradation [51]. Previous observations have demonstrated that Chl *a* is more salt-sensitive than Chl *b* and the primary reason for the decrease in the Chl content after salt stress is the degradation of Chl *a* [28]. The other reason for the reduction in the Chl content could be the decreased biosynthesis of the LHCII chlorophylls [46,72], which corresponds with an increase in the Chl *a/b* ratio and a decrease in the degree of thylakoid stacking [53]. The increase in the Chl *a/b* ratio was also reported in a salt- resistant rice variety [73] and in wheat plants [74]

The short time of salt treatment with the higher NaCl concentration led also to a decrease in the Car content, which is more pronounced in the EE than the TF line (Figure 2). These pigments can act as antioxidants and protect thylakoid membranes from oxidative stress in conditions of high salt content [75]. Previous studies with beans, cotton [76], rice [77], *Arabidopsis thaliana* [78], tobacco [79], barley [80], sugarcane [81], and chili peppers [82] revealed that the degree of carotenoid decrease depends on the salt sensitivity of plant species. Based on the above statements, a more effective defense and adaptation of the TF line could be assumed in comparison to the EE line to greater salt concentrations in the nutrient solution, i.e., the TF line is more resistant than the EE line. 

Additionally, the salt-induced structural changes and the modification of the pigment–protein complexes after 10 days of treatment with NaCl influenced the energy transfer between the pigment–protein complexes (Table 1). An increase in energy transfer from PSII to PSI was found (i.e., the F_735_/F_685_ ratio increased after Chl *a* or Chl *b* excitation), as this effect was greater in the TF line compared to the EE line. The increase in the energy transfer to PSI can be associated with increased lateral mobility of LHCII, as a result of salt-induced changes. This statement is confirmed by results obtained from previous studies, which revealed that the salinity induces uncoupling of the thylakoid membranes and/or an increase in the size of the PSI antenna [83,84,85], as well as the modification of the LHCII under salt stress [29,30]. Experimental results in this study revealed a salt-induced increase in the fluorescence emitted from the PSI core and PSI antenna in both studied lines. In addition, data also showed that LHCII^A^ and the ratio of the fluorescence emitted from LHCII^A^ to LHCII^M+T^ increased in both studied lines (Table 2). Previous studies showed an increase in aggregation of the LHCII under heat stress and high light intensity [86,87]. It has been suggested that aggregation of LHCII is a protective mechanism and it is connected with the dissipation of excess light energy [87]. In support of this assumption are our previous studies showing an increase in non-photochemical quenching in EE and TF lines after NaCl treatment [88]. 

Comparing the changes in the F_735_/F_685_ ratio upon excitation of Chl *a* (436 nm) and Chl *b* (472 nm), it could be assumed that under these conditions the energy transfer between Chl *b* and Chl *a* is affected. It has been suggested that the redistribution of absorbed excitation energy between the two photosystems is connected with the adaptation of the plants [89]. It can be proposed that the established increase in the energy transfer from PSII to PSI under salt treatment provides better protection for the TF line than the EE line.

In addition, data also revealed a salt-induced influence in the ratio F_695_/F_685_, indicating alterations in the energy transfer between the pigments in the PSII complex (Table 1) as a result from a modification of the PSII complex [9,27,29,30,34]. The changes in this ratio were accompanied by a decrease in the fluorescence from the PSII reaction center and PSII core antenna (Table 2). These changes in energy transfer could be a result of the salt-induced structural heterogeneity (alterations) of the PSII complex [25]. The influence on the energy transfer between pigment–protein complexes of the PSII has also been shown after salt treatment of wheat plants [36].

The salt-induced changes in pigment composition (Figure 1 and Figure 2) and energy transfer (Table 1) influence the PSII photochemical activity after short treatment (10 days) with 150 mM NaCl (Figure 4). The inhibition of the PSII-mediated electron transport could be a result of the altered structure of thylakoid membranes [90], modification of the pigment–protein complexes, an inhibition of the electron transport from Q_A_ to Q_B_, and an increase in Q_B_-non-reducing centers [7,25,27,32,91]. Previous investigations with wheat plants also showed an inhibition of PSII photochemistry under salt stress, as a result of modification of the donor and acceptor side of PSII [46]. The inhibition after treatment with higher NaCl in the EE line was greater than in the TF line (Figure 4). This assumption is also confirmed by our previous in vivo studies using PAM chlorophyll fluorescence [88]. It has been shown that under the same conditions (10 days and 150 mM NaCl) the photochemical quenching (qP) and photochemical energy conversion decrease, but non-photochemical quenching (qN) increases [88]. Experimental results also showed stronger inhibition of flash-induced oxygen yields (*Y*_3_) than the oxygen evolution under continuous illumination (amplitude A of oxygen burst) after 10 days of NaCl treatment (Figure 5), which suggests greater inactivation of PSIIα than PSIIβ centers [66,67]. In addition, our current results also revealed a reduction in PSI activity, which could be caused by the modification of the PsaB core subunit of PSI [36]. The inhibition of the photochemical activities of both photosystems after short treatment (10 days) with 150 mM NaCl (Figure 4), which correspond with a stronger increase in the flavonoids and proline [88], was greater in the TF line than the EE line [88]. It has been suggested that proline protects plants against abiotic stress, and its strong increase is one of the reasons for reversibility of the salt-induced changes in thylakoid membranes [92]. 

Data for the kinetic parameters of oxygen evolution after the flash and continuous illumination gives additional information about the influence of salt concentration on oxygen-evolving reactions (Figure 5 and Table 3). It has been found that the short treatment with a higher NaCl concentration influenced the initial S_0_-S_1_ state distribution of PSII centers (i.e., S_0_ increase). It is known that the oxidation state of the Mn clusters in S_0_ (Mn^2+^, Mn^3+^, Mn^4+^, Mn^4+^) is lower by one oxidizing equivalent than in the S_1_ state (Mn^3+^, Mn^3+^, Mn^4+^, Mn^4+^) [93]. A similar influence, on the PSII centers in the S_0_ state in dark-adapted samples, has also been shown after changes in the number and organization of the PSII-antenna complexes in pea plants [66]. 

At the same time, misses (α), double hits (β) and blocked centers (S_B_) increased, and this increase was more pronounced in the EE line than in the TF line (Table 3). It could be suggested that different effects of salinity on the kinetic parameters of oxygen evolution are due to the differences in the modification of the Mn cluster on the donor side [94], which is one reason for the different salt sensitivity of the studied *Paulownia* lines. The decrease in A_F_/A_S_ ratio after salt treatment revealed that the ratio of the functional active PSIIα (characterized by amplitude A_F_) to PSIIβ centers (characterized by amplitude A_S_) decreased. This statement is also supported by previously observed salt-induced changes in the PSII antenna size heterogeneity [20,25]. The impact on the kinetics of the initial oxygen burst decay after salt treatment might be due to the changes in the acceptor side of PSII [25,46] and the influence of the interaction between Q_B_ and plastoquinone [53,62], as well as the modification of the OEC [94,95]. Previously, changes were shown in the kinetic parameters of oxygen-evolving reactions after modifications of the pigment–protein complexes of the photosynthetic apparatus [67,96,97]. It has been shown fluridone treatment of peas is accompanied by a decrease in the functional active PSIIα to PSIIβ centers, i.e., the A_F_/A_S_ ratio decreases and there is an increase in misses (α) and double hits (β). The study of the influence of UV radiation and high light intensity on peas revealed an influence of the S_0_-S_1_ distribution and a decrease in the A_F_/A_S_ ratio. Previous studies also revealed that the changes in the organization of the PSII complex also influenced the kinetic parameters of oxygen-evolving reactions [67,96,97].

The current experimental results showed that after prolonged treatment with 100 mM NaCl the energy transfer between pigment–protein complexes, the photochemical activities of both photosystems, as well as the kinetic parameters of oxygen evolution, were similar to the corresponding controls in both studied lines. The values of the parameters studied above for TF treated with 150 mM NaCl were also similar to those of untreated plants, while this concentration was lethal for the EE line. Our previous in vivo studies of *Paulownia* plants, grown in saline soils for longer periods, also revealed better tolerance of the TF line than the EE line [53,62]. In addition, data showed that the rate of photosynthesis in the TF line increases in moderately saline soils, but in the EE line the rate decreases with increasing soil salinity [53,62]. All these observations suggest that the TF line is more tolerant to higher NaCl concentrations and adapts better after prolonged treatment.

## 4. Materials and Methods

### 4.1. Growth Conditions and Experimental Setup 

The experiments were carried out with seedlings of *Paulownia tomentosa* x *fortunei* and *Paulownia elongata* x *elongata*. The seedlings were received from “BioTree” Ltd., Sofia, Bulgaria. They were cultivated for 60 days in a vegetation photothermostat chamber with controlled conditions: 16/8 h photoperiod, light intensity of 200 μmoles photons m^−2^ s^−1^, 28/25 °C day/night temperature and relative humidity 50–75%. The plants were grown for 20 days in ¼ Hoagland solution and 40 days in ½ Hoagland solution containing: 2.5 mM KNO_3_, 2.5 mM Ca (NO_3_)_2_, 1 mM MgSO_4_, 0.5 mM NH_4_NO_3_, 0.5 mM K_2_HPO_4_, 23 μM H_3_BO_3_, 4.5 μM MnCl_2_, 0.4 μM ZnSO_4_, 0.2 μM CuSO_4_, 0.25 μM Na_2_MoO_4_ and 20 μM Fe-EDTA (pH 6.0). The solutions were aerated every day and were changed every three days. The treatment with two concentrations of NaCl (100 mM and 150 mM) was carried out on the 60- days-old plants. The measurements were made after short (10 days) and prolonged (25 days) treatment with NaCl. These variants were chosen based on our previous research with *Paulownia* grown on soils with different salinity [53,62]. Two independent experiments were performed with 4 replicates per variant. The third and fourth fully developed leaves were used for measurements. The plants of both lines after 10 days of treatment with different NaCl concentrations are shown in Appendix A.

### 4.2. Isolation of Thylakoid Membranes 

Thylakoid membranes were isolated from leaves as described in Harrison and Melis [98], with some modifications having in mind the specific features of the leaves of *Paulownia* plants. Fresh leaves were ground in a buffer containing: 50 mM Tricine-NaOH (pH 7.8), 0.4 M sucrose, 15 mM MgCl_2_, 10 mM NaCl, 50 mM sodium ascorbate, and 0.5% PVP. The homogenate was centrifuged at 2000× *g* for 2 min. Then, the supernatant was centrifuged at 5400× *g* for 15 min. The pellet was suspended in a buffer containing: 50 mM Tricine-NaOH (pH 7.8), 10 mM MgCl_2_, 10 mM NaCl (pH 7.8) and the suspension was centrifuged again for 12 min at 12,000× *g*. The pellet was resuspended in a buffer for respective measurements.

### 4.3. Absorption Spectra 

The absorption spectra of thylakoid membranes were recorded in a buffer containing: 40 mM HEPES (pH 7.6), 5 mM MgCl_2_, 10 mM NaCl and 0.4 M sucrose. The chlorophyll concentration was 10 μg Chl ml^−1^. The measurements were made on a spectrometer Specord 210 Plus., Analytic Jena, Jena, Germany.

### 4.4. Pigment Analysis 

The determination of pigment content was performed as described in Stefanov et al. [6]. The pigments were extracted from leaves with an ice-cold 80% (*v*/*v*) acetone in the dark. The resulting homogenates were centrifuged at 2500× *g* for 8 min at 4 °C. The amount of pigments (Chl *a*, Chl *b*, and Car) in the supernatant were measured spectrophotometrically (with Specord 210 Plus, Analytic Jena, Jena, Germany) at 470 nm, 646.8 nm, and 663.2 nm, using the equations of Lichtenthaler [99].

### 4.5. Low-Temperature Fluorescence Measurements

Thylakoid membranes were suspended in a buffer containing 40 mM HEPES (pH 7.6), 10 mM NaCl, 5 mM MgCl_2_ and 0.4 M sucrose. The samples were frozen in liquid nitrogen and measured using a spectrofluorometer Jobin Yvon JY3 (Division Instruments S.A., Longjumeau, France) equipped with a red-sensitive photomultiplier and a low-temperature device. The fluorescence emission spectra were measured after excitation with 436 nm (for Chl *a*) or/and 472 nm (for Chl *b*). We calculated the chlorophyll emission ratios F_735_/F_685_ (for estimation of the energy redistribution between the two photosystems) and F_695_/F_685_ (for the energy transfer between chlorophyll–protein complexes in the LHCII–PSII supercomplex [64]. Gaussian decomposition of the fluorescence emission spectra was made as in Andreeva et al. [65].

### 4.6. Photochemical Activity of PSI and PSII

The photochemical activities of the two photosystems measured as PSI- and PSII-mediated electron transport were determined polarographically with a Clark-type oxygen electrode (Model DW1, Hansatech, Instruments Ltd., Norfolk, UK). The photochemical activity of PSII (H_2_O → BQ) was determined in a reaction medium containing: 20 mM MES-NaOH (pH 6.5), 5 mM MgCl_2_, 10 mM NaCl, 0.4 M sucrose, and 40 mM BQ (1–4 benzoquinone). The reaction for PSI activity (DCPIPH_2_ → MV) was estimated in a reaction medium containing: 50 mM Tricine (pH 7.8), 5 mM MgCl_2_, 10 mM NaCl, 0.4 M sucrose, 6 mM NH_4_Cl, 0.1 mM DCMU, 0.5 mM DCPIPH_2_, 10 mM Na ascorbate, and 0.1 mM MV. The chlorophyll concentration was 25 µg Chl.mL^−1^. 

### 4.7. Measurement of Oxygen Evolution under Flash and Continuous Illumination

The flash-induced oxygen yields and initial oxygen burst under continuous illumination were determined on isolated thylakoid membranes, using a polarographic oxygen rate electrode (Joliot-type) described by Zeinalov [100]. The measurements were made without artificial electron acceptors. Thylakoid membranes were suspended in a buffer containing: 40 mM HEPES (pH 7.6), 10 mM NaCl, 5 mM MgCl_2_, and 0.4 M sucrose. The chlorophyll concentration was 150 µg Chl.mL^−1^. The flash oxygen yields were induced by saturating (4 J) and short (T_½_ = 10 µs) periodic flash light sequences as in Ivanova et al. [67]. The following parameters, obtained from the model of Kok [100,101], were estimated: percentage of oxygen-evolving PSII centers in the most reduced (S_0_) state in the dark, misses (α), and double hits (β). The blocked oxygen-evolving PSII centers (S_B_) were calculated, using an extended kinetic version of Kok’s model [for details see Zeinalov [102]. The maximum amplitudes of flash-induced oxygen yields observed after the third flash (*Y_3_*) were used to assess the inhibition of the active PSII centers evolving oxygen by a non-cooperative mechanism (PSIIα in grana domains).

The oxygen evolution under continuous light irradiation with an intensity of 400 µmoles photons m^−2^ s^−1^ was measured to estimate the parameter *A* (amplitude of the initial oxygen burst). The oxygen burst decay of the induction curves can be deconvoluted with two exponents (fast and slow) with two amplitudes (A_F_ and A_S_) as described in Ivanova et al. [67]. The ratio A_F_/A_S_ is proposed to correspond to the proportion of the fast-operating centers evolving oxygen by non-cooperative (A_F_) and slow-operating centers (A_S_) by cooperative mechanisms, respectively, i.e., to the ratio of the functionally active PSIIα in grana to functionally inactive PSIIβ centers in stroma lamellae [66]. A typical signal of the oxygen evolution under continuous illumination and biphasic kinetics of oxygen burst decay are shown in Appendix A.

### 4.8. Statistical Analysis

Data were presented as mean values (± SE). The means were calculated from at least two independent experiments with four replicates of each variant. Statistically significant differences between variants of studied parameters were identified by analysis of variance (ANOVA), and a Tukey’s post hoc test was performed for each parameter to find statistically significant changes. Values of *p* < 0.05 were considered as significantly different. 

## 5. Conclusions

In summary, data revealed different effects of salinity on the function of the photosynthetic apparatus of studied *Paulownia* lines depending on the NaCl concentrations and the time of treatment. Short treatment (10 days) with a higher NaCl concentration (150 mM NaCl) led to: (i) a decrease in the amount of Chl *a* and Chl *b*; (ii) an influence on the energy transfer between pigment–protein complexes of PSII and on the redistribution of the excitation energy between PSII and PSI; (iii) an inhibition of the photochemical activities of PSII and PSI; and (iv) an influence on the kinetic parameters of the oxygen-evolving reactions. The changes in these parameters are more strongly pronounced in the EE line than in the TF line. Moreover, the TF line was adapted to the higher NaCl concentration after prolonged treatment. The better salt tolerance of the TF line than the EE line determined the former as more suitable for cultivation and phytoremediation of saline soils.

## Figures and Tables

**Figure 1 ijms-24-03108-f001:**
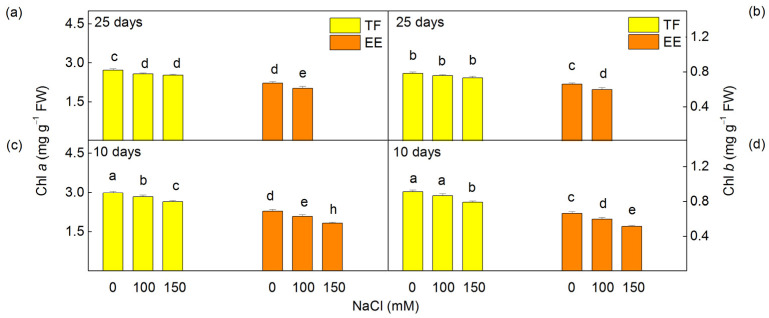
Effect of different concentrations of NaCl on the chlorophyll content of *Paulownia tomentosa* x *fortunei* (TF) and *Paulownia elongata* x *elongata* (EE) leaves. The plants were treated for 10 (**c**,**d**) and 25 days (**a**,**b**). Mean values (±SE) for a given parameter marked with different letters are statistically significant at *p* < 0.05.

**Figure 2 ijms-24-03108-f002:**
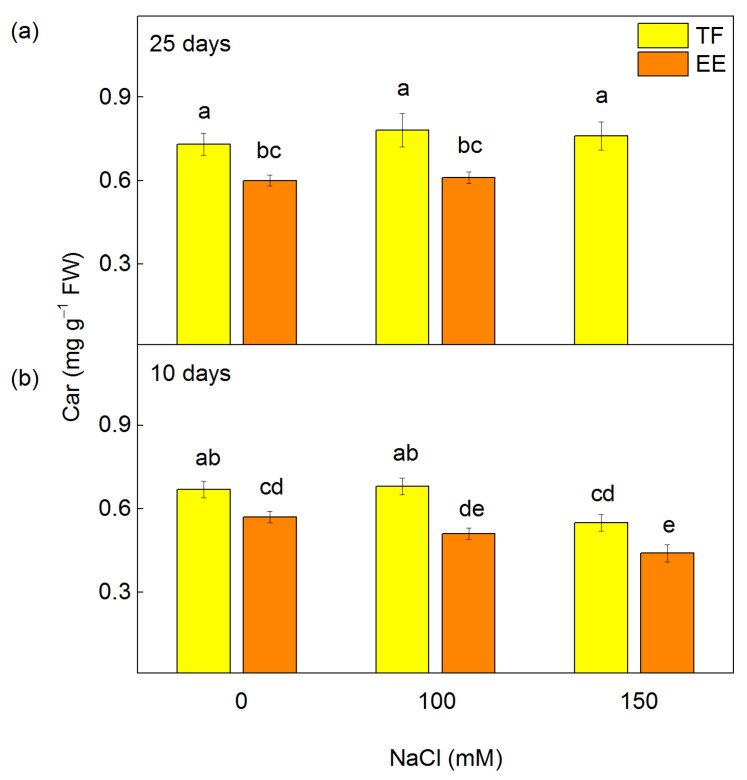
Effect of different concentrations of NaCl on the carotenoid content in leaves of *Paulownia tomentosa* x *fortunei* (TF) and *Paulownia elongata* x *elongata* (EE). The plants were treated for 10 (**b**) and 25 days (**a**). Mean values (±SE) with different letters have statistically significant differences at *p* < 0.05.

**Figure 3 ijms-24-03108-f003:**
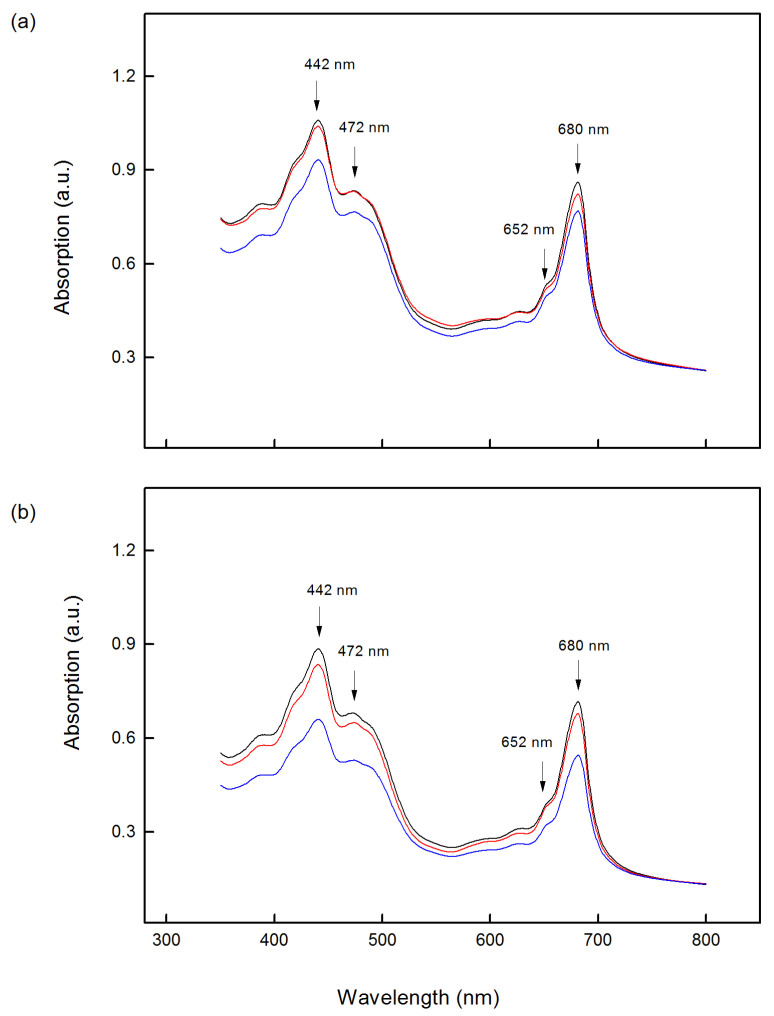
Absorption spectra of thylakoid membranes isolated from the leaves of *Paulownia tomentosa* x *fortunei* (TF) (**a**) and *Paulownia elongata* x *elongata* (EE) (**b**) treated for 10 days with different concentrations of NaCl; black curve (control), red curve (treatment with 100 mM NaCl) and blue curve (treatment with 150 mM NaCl).

**Figure 4 ijms-24-03108-f004:**
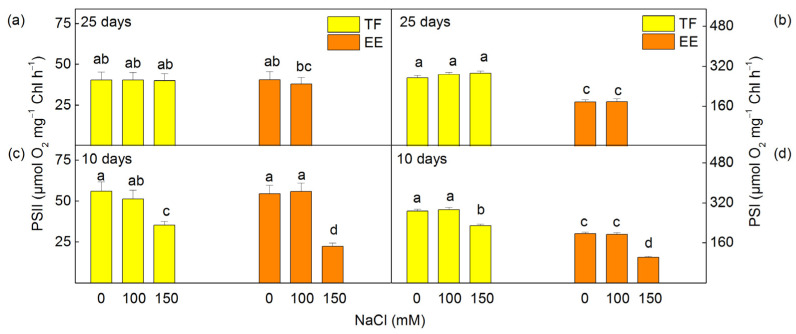
Influence of different NaCl concentrations on the PSII activity (H_2_O → BQ) (**a**,**c**) and PSI activity (DCPIPH_2_ → MV) (**b**,**d**) in isolated thylakoid membranes from leaves of *Paulownia tomentosa* x *fortunei* (TF) and *Paulownia elongata* x *elongata* (EE). The plants were treated after 10 and 25 days. Mean values (±SE) for the given parameter marked with different letters have statistically significant differences at *p* < 0.05.

**Figure 5 ijms-24-03108-f005:**
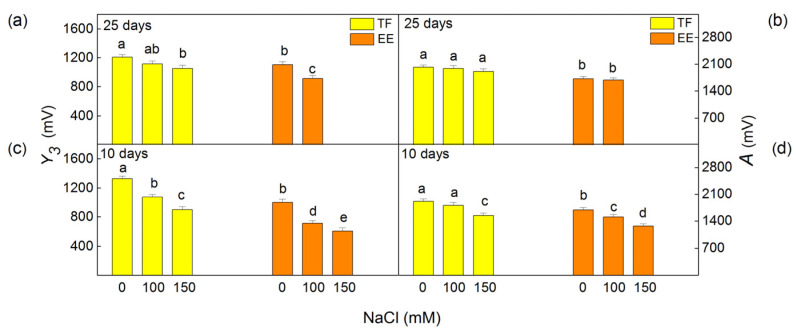
Influence of different NaCl concentrations on the maximum flash-induced oxygen yields (*Y_3_*) (**a**,**c**) and the amplitudes of initial oxygen burst under continuous illumination (*A*) (**b**,**d**) of isolated thylakoid membranes from leaves of *Paulownia tomentosa* x *fortunei* (TF) and *Paulownia elongata* x *elongata* (EE). The plants were treated for 10 and 25 days. Mean values (±SE) for each parameter marked with different letters have statistically significant differences at *p* < 0.05.

**Table 1 ijms-24-03108-t001:** Influence of different NaCl concentrations on the low-temperature (77 K) fluorescence emission ratios(F_695_/F_685_ and F_735_/F_685_)of isolated thylakoid membranes from leaves of *Paulownia tomentosa* x *fortunei* (TF) and *Paulownia elongata* x *elongata* (EE). The plants were treated for 10 and 25 days. The thylakoid membranes were excited with 436 nm and 472 nm. Means values (±SE) in each column marked with different letters have statistically significant differences at *p* < 0.05. *—lethal NaCl concentration.

	NaCl	λexc = 436 nm		λexc = 472 nm	
	(mM)	F_735_/F_685_	F_695_/F_685_	F_735_/F_685_	F_695_/F_685_
10 days					
TF	0	1.553 ± 0.031 ^d^	0.854 ± 0.017 ^c,d^	1.257 ± 0.026 ^f^	0.934 ± 0.014 ^b^
	100	1.570 ± 0.035 ^d^	0.882 ± 0.013 ^c,d^	1.251 ± 0.030 ^f^	0.921 ± 0.007 ^b^
	150	1.784 ± 0.029 ^b,c^	0.933 ± 0.004 ^b^	1.332 ± 0.018 ^d,e^	0.951 ± 0.016 ^a^
EE	0	1.715 ± 0.029 ^c^	0.924 ± 0.010 ^b^	1.442 ± 0.031 ^c^	0.909 ± 0.025 ^b^
	100	1.715 ± 0.025 ^c^	0.923 ± 0.009 ^b^	1.410 ± 0.017 ^c^	0.898 ± 0.022 ^b^
	150	1.798 ± 0.019 ^b^	0.995 ± 0.016 ^a^	1.682 ± 0.044 ^a^	0.947 ± 0.023 ^a^
25 days					
TF	0	1.340 ± 0.019 ^e^	0.842 ± 0.016 ^d^	1.225± 0.025 ^f^	0.847 ± 0.014 ^c^
	100	1.347 ± 0.011 ^e^	0.867 ± 0.017 ^c,d^	1.276 ± 0.021 ^e,f^	0.850 ± 0.017 ^c^
	150	1.427 ± 0.039 ^e^	0.869 ± 0.008 ^c,d^	1.292 ± 0.023 ^e,f^	0.880 ± 0.013 ^b,c^
EE	0	1.909 ± 0.039 ^a^	0.884 ± 0.014 ^c,d^	1.484 ± 0.026 ^b,c^	0.944 ± 0.022 ^a^
	100	1.954 ± 0.041 ^a^	0.886 ± 0.013 ^c^	1.538 ± 0.030 ^b^	0.957 ± 0.030 ^a^
	150	*	*	*	*

**Table 2 ijms-24-03108-t002:** Fluorescence emission from the pigment–protein complexes in thylakoid membranes of *Paulownia tomentosa* x *fortunei* (TF) and *Paulownia elongata* x *elongata* (EE) treated for 10 days with different NaCl concentrations: fluorescence emitted from monomers (^M^) and trimers (^T^) of LHCII (LHCII^M+T^, F_680_), PSII reaction center (PSII RC, F_685_); PSII antenna (F_695_), aggregated (^A^) LHCII (LHCII^A^, F_700_), PSI core (F_720_) and PSI antenna (F_735_). The thylakoid membranes were excited with 436 nm. The area was calculated as % from the total area of emission spectra. Means values (±SE) in each column marked with different letters have statistically significant differences at *p* < 0.05.

NaCl	Area (%)					
(mM)	LHCII^M+T^	PSII RC	PSII Antenna	LHCII^A^	PSI Core	PSI Antenna
TF						
0	9.161 ± 0.067 ^a^	21.642 ± 0.131 ^a^	20.676 ± 0.109 ^a^	8.995 ± 0.087 ^e^	12.128 ± 0.065 ^e^	27.399 ± 0.044 ^d^
100	9.184 ± 0.057 ^a^	18.925 ± 0.115 ^c^	17.945 ± 0.095 ^c^	10.697 ± 0.104 ^b,c^	14.267 ± 0.0786 ^c^	28.981 ± 0.046 ^c^
150	7.595 ± 0.056 ^b^	17.704 ± 0.129 ^d^	17.251 ± 0.105 ^d^	10.443 ± 0.055 ^b,c^	17.4362 ± 0.170 ^a^	29.571 ± 0.160 b
EE						
0	7.047 ± 0.085 ^c^	20.4834 ± 0.151 ^b^	19.083 ± 0.036 ^b^	10.735 ± 0.096 ^b^	13.612 ± 0.122 ^d^	29.039 ± 0.285 ^b,c^
100	7.078 ± 0.045 ^c^	18.978 ± 0.048 ^c^	19.075 ± 0.067 ^b^	12.389 ± 0.010 ^a^	13.666 ± 0.035 ^d^	28.812 ± 0.150 ^c^
150	6.054 ± 0.070 ^d^	16.319 ± 0.039 ^e^	16.319 ± 0.119 ^d^	14.455 ± 0.098 ^d^	14.984 ± 0.096 ^b^	34.518 ± 0.078 ^a^

**Table 3 ijms-24-03108-t003:** Influence of different concentrations of NaCl on the kinetic parameters of oxygen evolution of isolated thylakoid membranes from *Paulownia tomentosa* x *fortunei* (TF) and *Paulownia elongata* x *elongata* (EE); S_0_—the percentage of PSII centers in S_0_ state in the dark; α—misses, β—double hits, S_B_—the blocked PSII centers and A_F_/A_S_—the ratio of fast (PSII α) to slow (PSII β) oxygen-evolving PSII centers. The plants were treated with NaCl for 10 and 25 days. Means values (±SE) in each column marked with different letters have statistically significant differences at *p* < 0.05. *—lethal NaCl concentration.

	NaCl (mM)	S_0_ (%)	α (%)	β (%)	S_B_	A_F_/A_S_
10 days						
TF	0	24.70 ± 1.04 ^c^	22.96 ± 1.17 ^c^	5.78 ± 0.36 ^c^	0.98.± 0.11 ^d^	2.88 ± 0.17 ^a^
	100	23.41 ± 0.85 ^c^	24.81 ± 1.01 ^b,c^	5 27 ± 0.30 ^c^	1.02 ± 0.05 ^d^	2.59 ± 0.14 ^a,b^
	150	30.73 ± 1.78 ^a^	27.42 ± 1.05 ^b^	7.64 ± 0.44 ^ab^	1.39 ± 0.08 ^b^	1.87 ± 0.18 ^c^
EE	0	24.37 ± 0.55 ^c^	23.29 ± 1.14 ^b,c^	5.29 ± 0.33 ^c^	1.11 ± 0.10 ^cd^	2.43 ± 0.16 ^b^
	100	26.58 ± 1.12 ^b,c^	24.40 ± 1.18 ^b^	5.54 ± 0.33 ^c^	1.34 ± 0.09 ^b^	2.32 ± 0.14 ^b^
	150	31.93 ± 1.49 ^a^	29.44 ± 1.02 ^a^	8.47 ± 0.52 ^a^	1.73 ± 0.11 ^a^	1.93 ± 0.15 ^c^
25 days						
TF	0	23.80 ± 0.68 ^c^	24.32 ± 0.86 ^c^	4.53 ± 0.33 ^c^	1.15 ± 0.09 ^c^	2.67± 0.16 ^ab^
	100	22.35 ± 1.12 ^c^	25.19 ± 0.82 ^b,c^	4.19 ± 0.21 ^c^	1.25 ± 0.11 ^c,b^	2.56 ± 0.07 ^b^
	150	26.41 ± 1.03 ^b^	27.58 ± 0.99 ^b^	6.08 ± 0.80 ^b^	1.33 ± 0.10 ^c,b^	2.42 ± 0.13 ^b^
EE	0	23.72 ± 0.63 ^c^	26.40 ± 1.08 ^b^	5.92 ± 0.23 ^c^	1.19 ± 0.11 ^c^	2.39± 0.15 ^b^
	100	25.51 ± 0.69 ^c^	27.53 ± 1.02 ^b^	5.97 ± 0.22 ^c^	1.38 ± 0.13 ^c,b^	2.34 ± 0.17 ^b^
	150	*	*	*	*	*

## Data Availability

Not applicable.

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
