# Peer review of "Impact of Salinity on the Energy Transfer between Pigment–Protein Complexes in Photosynthetic Apparatus, Functions of the Oxygen-Evolving Complex and Photochemical Activities of Photosystem II and Photosystem I in Two Paulownia Lines"

_ijms, 2023, doi:10.3390/ijms24043108_

Round 1
Reviewer 1 Report
Dear Authors,
Thank you for putting effort into understanding salinity's effects on photosynthetic apparatus in plants. This is a concern of the present day and future that must be addressed.
This paper uses some really heavy structures and it wasn't an easy read. I would suggest authors review this paper thoroughly and re-write some parts of the result and discussion section in a way that it's easier for the readers to understand it. Please address the disconnect in your findings. Providing some more details on the background of each phenomenon might help.
Please mention clearly that you have used a spectrophotometer for some measurements in methods and results.
I have the following questions regarding the design of the experiment:
Did you perform only spectrophotometer-based measurements to analyze salinity effects on pigments? I would suggest using other analytical techniques to make solid conclusions about your finding. This paper can greatly improve if you are able to include analytical measurements using commonly used chromatographic techniques (GCMS or LCMS).
Also when you separate the thylakoid membranes, did you perform any protein analysis or any other kind of study to highlight the differences in membrane proteins in response to salinity? This can be an easy experiment and add value to your findings.
I also think you can include the leaf images (optional) subjected to a saline environment compared to normal conditions.
I understand the whole train of thought here, but there is still so much room for improvement.
I highly encourage the authors to review this paper and resubmit it after addressing the reviewer's concerns.
Best Regards,
Author Response
Report to the comments of reviewer 1 on manuscript titled “Impact of Salinity on the Energy Transfer between Pigment-Protein Complexes in Photosynthetic Apparatus, Functions of the Oxygen-Evolving Complex and Photochemical Activities of Photosystem II and Photosystem I in Two Paulownia Lines” by Stefanov et al.
Dear Reviewer,
We would like to thank you for constructive and insightful comments in relation to this work. We considered all comments and suggestions to be justified, and corrected the manuscript accordingly. Please, find the detailed list of all edits below. The newly edited text parts are indicated with red letter.
Thank you for putting effort into understanding salinity's effects on photosynthetic apparatus in plants. This is a concern of the present day and future that must be addressed.This paper uses some really heavy structures and it wasn't an easy read. I would suggest authors review this paper thoroughly and re-write some parts of the result and discussion section in a way that it's easier for the readers to understand it. Please address the disconnect in your findings. Providing some more details on the background of each phenomenon might help.
Answer: Thanks for the useful comment. Some paragraphs of the Results and Discussion sections have been rewritten in the revised version of the MS.
Please mention clearly that you have used a spectrophotometer for some measurements in methods and results.
Answer: It was made in the revised MS.
I have the following questions regarding the design of the experiment:
Did you perform only spectrophotometer-based measurements to analyze salinity effects on pigments? I would suggest using other analytical techniques to make solid conclusions about your finding. This paper can greatly improve if you are able to include analytical measurements using commonly used chromatographic techniques (GCMS or LCMS). Also when you separate the thylakoid membranes, did you perform any protein analysis or any other kind of study to highlight the differences in membrane proteins in response to salinity? This can be an easy experiment and add value to your findings.
Answer: Unfortunately, we could not apply the proposed techniques for the analysis of pigments and proteins under salt stress conditions. Applying these methods means redoing the experiments. Ordering the sprouts and growing the plants will take us more than 6-8 months. To provide additional insight the effects of salinity on the pigment-protein complexes of the photosynthetic apparatus, the participation of individual complexes in the fluorescence spectrum has been determined - Table 2 in the revised manuscript.
I also think you can include the leaf images (optional) subjected to a saline environment compared to normal conditions.
Answer: Following your recommendation we have added Figure 3S.
I understand the whole train of thought here, but there is still so much room for improvement. I highly encourage the authors to review this paper and resubmit it after addressing the reviewer's concerns.
Answer: Taking into account your comments and recommendations, we hope that the revised version of the manuscript has been improved.
Sincerely yours,
Dr. Emilia Apostolova

Reviewer 2 Report
The manuscript entitled “Impact of Salinity on the Energy Transfer between Pigment-Protein Complexes in Photosynthetic Apparatus, Functions of the Oxygen-Evolving Complex and Photochemical Activities of Photosystem II and Photosystem I in Two Paulownia Lines” intended for publication in International Journal of Molecular Sciences, Special Issue: Molecular Mechanisms of Plant Defense against Abiotic Stress is generally an interesting paper, however I think that manuscript needs improvements before publication.
I have mainly minor remarks. Generally, the paper is relatively straightforward and well written, however some parts of manuscript need more attention. The Authors could more specify the purpose of the study. In addition, Authors could improve Material and methods section, provide more information on plant material and pigment extractions. I think, the Authors should move Table 2 to Result section (from Discussion). Discussion should be longer and and the possibilities of using more tolerant paulownias on saline soils should be indicated. The Authors should check more carefully the Reference list, and improve it. In addition, there are many small mistakes in the text of manuscript, including Reference list, that need to be corrected by Authors (e.g. lines: 22, 187, 276, 309, 378, 398, 477, 480, 488, 503, 517, 526, 542, 543, 555, 563. 571, 576, 578, 584, 598, 601, 621, 634, 640, 649, 655, 657, 672).
Author Response
Report to the comments of reviewer 1 on manuscript titled “Impact of Salinity on the Energy Transfer between Pigment-Protein Complexes in Photosynthetic Apparatus, Functions of the Oxygen-Evolving Complex and Photochemical Activities of Photosystem II and Photosystem I in Two Paulownia Lines” by Stefanov et al.
Dear Reviewer,
We would like to thank you for constructive and insightful comments in relation to this work. We considered all comments and suggestions to be justified, and corrected the manuscript accordingly. Please, find the detailed list of all edits below. The newly edited text parts are indicated with red letter.
Comments and Suggestions for Authors
The manuscript entitled “Impact of Salinity on the Energy Transfer between Pigment-Protein Complexes in Photosynthetic Apparatus, Functions of the Oxygen-Evolving Complex and Photochemical Activities of Photosystem II and Photosystem I in Two Paulownia Lines” intended for publication in International Journal of Molecular Sciences, Special Issue: Molecular Mechanisms of Plant Defense against Abiotic Stress is generally an interesting paper, however I think that manuscript needs improvements before publication.
I have mainly minor remarks. Generally, the paper is relatively straightforward and well written, however some parts of manuscript need more attention. The Authors could more specify the purpose of the study. In addition, Authors could improve Material and methods section, provide more information on plant material and pigment extractions. I think, the Authors should move Table 2 to Result section (from Discussion). Discussion should be longer and and the possibilities of using more tolerant paulownias on saline soils should be indicated. The Authors should check more carefully the Reference list, and improve it. In addition, there are many small mistakes in the text of manuscript, including Reference list, that need to be corrected by Authors (e.g. lines: 22, 187, 276, 309, 378, 398, 477, 480, 488, 503, 517, 526, 542, 543, 555, 563. 571, 576, 578, 584, 598, 601, 621, 634, 640, 649, 655, 657, 672).
Answer: Thanks for the helpful comments. All these suggestions and remarks are taken into account in the revised version of the MS. We changed some paragraphs in the manuscript (Introduction, Materials and Methods, Results, and Discussion) which are marked in red. Reference corrections have also been made and new references were added in the revised manuscript.
Table 2 has been added to the revised manuscript at the suggestion of the other reviewer. One figure (Figure 3S) in the Supplementary Material has also been added.
Sincerely yours,
Dr. Emilia Apostolova

Round 2
Reviewer 1 Report
Dear Authors,
Thank you for making these changes. These details were crucial and I am sure these will help the readers understand your research better.
Regards